# Impact of Environmental Exposure on Chronic Diseases in China and Assessment of Population Health Vulnerability

**Zhibin Huang** [1,2] , **Chunxiang Cao** [1,3,*], **Min Xu** [1] **and Xinwei Yang** [1]

1    State Key Laboratory of Remote Sensing Science, Aerospace Information Research Institute,
     Chinese Academy of Sciences, Beijing 100101, China
2    University of Chinese Academy of Sciences, Beijing 100094, China
3    Canter for Application of Spatial IT in Public Health, Chinese Academy of Sciences, Beijing 100101, China
*    Correspondence: caocx@aircas.ac.cn; Tel.: +86-010-6483-6205

**Abstract:** Although numerous epidemiological studies have demonstrated a relationship between environmental factors and chronic diseases, there is a lack of comprehensive population health vulnerability assessment studies from the perspective of environmental exposure, population sensitivity and adaptation on a regional scale. To address this gap, this study focused on six high-mortality chronic diseases in China and constructed an exposure–sensitivity–adaptability framework-based index system using multivariate data. The constructed system effectively estimated health vulnerability for the chronic diseases. The R-square between vulnerability and mortality rates for respiratory diseases and malignant tumors exceeded 0.7 and was around 0.6 for the other four chronic diseases. In 2020, Chongqing exhibited the highest vulnerability to respiratory diseases. For heart diseases, vulnerability values exceeding 0.5 were observed mainly in northern and northeastern provinces. Vulnerability values above 0.5 were observed in Jiangsu, Shanghai, Tianjin, Shandong and Liaoning for cerebrovascular diseases and malignant tumors. Shanghai had the highest vulnerability to endogenous metabolic diseases, and Tibet exhibited the highest vulnerability to digestive system diseases. The main related factor analysis results show that high temperature and humidity, severe temperature fluctuations, serious air pollution, high proportion of middle-aged and elderly population, as well as high consumption of aquatic products, red meat and eggs increased health vulnerability, while increasing per capita educational resources helped reduce vulnerability.

**Keywords:** exposure-sensitivity-adaptability framework; population health vulnerability assessment; remote sensing; chronic diseases; China

## 1. Introduction

China has undergone rapid urbanization in recent years, with the urbanization rate of the resident population increasing from 36.09% in 2000 to 49.68% in 2010 to 63.89% in 2020. However, this growth has come at a cost. The environmental impact of this rapid urbanization has led to fragmentation of natural habitat, reduction in biodiversity, water pollution, urban heat islands and frequent haze. These issues have negatively affected the physical and mental health, as well as the quality of life of urban residents [1–7]. Additionally, compared with developed countries, the health of residents living in developing countries seems to be more vulnerable to the deterioration of urban environmental quality.

Chronic diseases mainly include malignant tumors, cardiovascular and cerebrovascular diseases, heart diseases, hypertension, diabetes, mental illnesses and a series of non-infectious diseases which cannot be recovered by the patients themselves. With the development of the social economy and changes in people's lifestyle, the disease spectrum of Chinese people has undergone tremendous changes, and the prevalence of chronic diseases closely related to environmental factors and unhealthy lifestyles has become increasingly serious. Chronic diseases have accounted for 87% of total deaths and 70% of the total disease burden in China, posing severe challenges for prevention and treatment [8]. The 2011

World Economic Risk Report warned that five major chronic diseases, mainly including cardiovascular diseases, tumors, diabetes, respiratory diseases and mental diseases, would have a profound impact on the country's medical system and economic system in the next two decades. Studying the factors related to the risk of chronic disease deaths is crucial due to the significant threat that chronic diseases pose to public health and the overall well-being of the country.

Numerous epidemiological studies have demonstrated the impact of meteorological factors and air pollution on the occurrence and development of chronic diseases. For example, the impact of high temperatures on the respiratory system and cardiovascular system will lead to an increase in thermal mortality [9,10]. Severe temperature changes, such as large daily temperature ranges, have significant adverse effects on human health and are a risk factor for acute stroke deaths [11–13]. Urbanization has been linked to air pollution [14–16], which was identified as the most substantial risk factor for increasing disease burden in 2013 and accounts for approximately 4 million deaths annually [17]. The association between exposure to multiple mixtures in the air including PM, $O_3$, $NO_2$ and premature death has become the focal point of many epidemiological studies [18–31].

On the one hand, these studies that focus on a single factor can only reflect one aspect of the environment, which does not allow for a comprehensive assessment of the vulnerability of people exposed to environmental factors. On the other hand, the description of environmental factors in most studies is not sufficiently detailed. For instance, the data collected from meteorological stations and air quality monitoring stations in many studies only represent a single location and its surrounding areas. Since the distribution of stations in most areas is not uniform, interpolation methods used to generate spatial data may result in uneven distribution pattern, particularly in areas where stations are sparse. Moreover, the cohort and sampling survey methods adopted by most epidemiological studies are time-consuming and laborious. Due to the high threshold of data acquisition, the research scope is limited [32], and there is a lack of research on the relationship between environmental factors and population health at a regional scale.

In this study, six types of chronic diseases with high mortality in China were taken as the research object and the evaluation indicators were selected based on the framework of exposure–sensitivity–adaptability. To address the issue of insufficient detail regarding environmental elements in previous studies, remote sensing products with high spatial and temporal resolution were applied to the exposure indexes calculation. In previous studies, literature review was primarily used to identify indicators for constructing the population health vulnerability assessment indicator system. However, there was limited analysis conducted on whether these indicators were correlated with population health and with each other. This limitation could potentially reduce the effectiveness of indicator selection and result in the portrayal of similar characteristics by multiple indicators in one aspect. Therefore, in this study, taking the mortality of the six types of chronic diseases in some provinces from 2010 to 2019 as response variables, the indicators were screened by geographic detector and correlation analysis, and the weights of the indicators were determined based on quantile regression to build provincial-scale population health vulnerability evaluation index systems for the six kinds of chronic diseases. The health vulnerability values in each province in 2020 were estimated by using the constructed index systems. To address the lack of verification of vulnerability assessment results in relevant vulnerability studies, this study verified the effectiveness of the constructed evaluation index systems by calculating the determination coefficient between vulnerability values and corresponding chronic disease mortality.

## 2. Materials and Methods

### 2.1. Study Area

The study area contains 31 provinces (including autonomous regions and municipalities) in the Chinese mainland region (Figure 1), including Heilongjiang, Jilin and Liaoning in northeast China; Beijing, Tianjin, Hebei, Shanxi and Inner Mongolia Autonomous Re-

gion in north China; Shaanxi, Gansu, Qinghai, Ningxia Hui Autonomous Region and Xinjiang Uygur Autonomous Region in northwest China; Shandong, Shanghai, Jiangsu, Zhejiang, Anhui, Fujian and Jiangxi in east China; Henan, Hubei and Hunan in central China; Guangdong, Guangxi Zhuang Autonomous Region and Hainan in south China; as well as Chongqing, Sichuan, Guizhou, Yunnan and Tibet Autonomous Region in southwest China. The 31 provinces have a wide geographical range, with latitudes of about 50° between north and south and longitude of nearly 62° between east and west. There are significant differences in climatic conditions and economic development among regions.

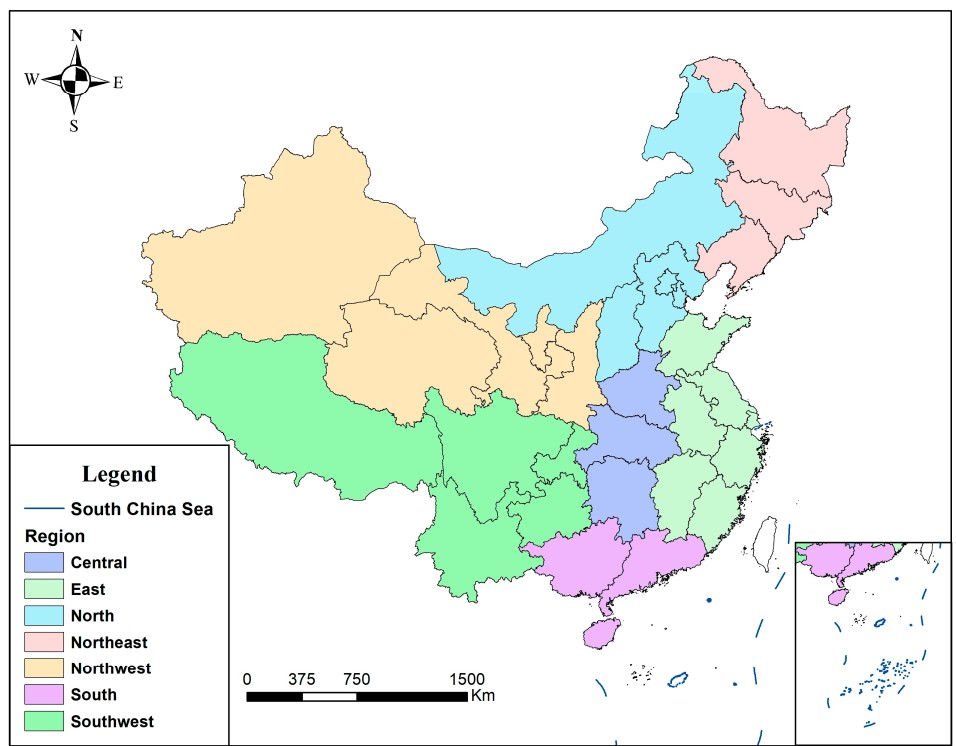

**Figure 1.** Regional division map of the study area.

## 2.2. Exposure–Sensitivity–Adaptability Framework

According to the definition of vulnerability in the Third Assessment Report of IPCC, vulnerability is a comprehensive evaluation standard for the system's susceptibility to the adverse effects of climate change, and a function of climate risk exposure level, system sensitivity and adaptability. This study extended the exposure to climate change to the exposure to environmental risk factors in this definition and sorted out the corresponding evaluation indicators from three aspects: exposure, sensitivity and adaptability (Table 1).

Exposure mainly refers to the characteristics and extent of systematic exposure to environmental risk factors, which determine the potential harm that environmental risk factors may cause to the population [33,34]. This study selected indicators to characterize exposure from two aspects of meteorology and air quality. Sensitivity mainly refers to the degree to which the system is affected by environmental risk factors. Due to variations in physical and economic conditions, different groups of people can withstand different impacts of environmental exposure. In this study, we selected indicators to characterize sensitivity mainly from five aspects: age and gender composition, education level, occupation, food consumption and land use. Adaptability mainly refers to the ability of the system to adapt to environmental risks, to reduce potential losses or to deal with the consequences of such risks [35]. This study focused on indicators to characterize adaptability from the perspective of regional economic foundation and infrastructure construction.

**Table 1.** Population health vulnerability evaluation indexes table.

| Indicator Dimensions | Indicator Meaning | Indicator Representation |
|---|---|---|
| **Exposure (36)** | nth percentile of daily average temperature (population weighted) | WLST_nth ($n$ = 1,3,10,25,50,75,90,97,99) |
| | Daily temperature standard deviation (population weighted) | WTSD |
| | Daily temperature difference nth percentile (population weighted) | WDTD_nth ($n$ = 10,25,50,75,90) |
| | Intraday–interday temperature variability nth percentile (population weighted) | WIITV_nth ($n$ = 10,25,50,75,90) |
| | Annual average $PM_{2.5}$/$PM_{10}$ concentration (population weighted) | WPM25_YM, WPM10_YM |
| | Days when the $PM_{2.5}$/$PM_{10}$ concentrations did not reach the guideline values for the three transitional periods formulated by WHO (population weighted) | WPM25_ITn ($n$ = 1,2,3) WPM10_ITn ($n$ = 1,2,3) |
| | Annual average $SO_2$ concentration | SO2_YM |
| | Annual average $NO_2$ concentration | NO2_YM |
| | Annual average CO 24-h mean 95th percentile concentration | CO_YM |
| | Annual average $O_3$ daily maximum 8 h moving average 90th percentile concentration | O3_YM |
| | Percentage of days with AQI not in good condition in a year | AQI |
| | Annual average relative humidity, wind speed, and surface pressure (population weighted) | WRH, WSP, WWS10 m |
| **Sensitivity (24)** | Proportion of population aged 0–4, 5–14, 15–19, 20–39, 40–59, 60–79 and over 80 | Age_0_4, Age_5_14, Age_15_19, Age_20_39, Age_40_59, Age_60_79, Age_80 |
| | Male/female sex ratio | MF_Ratio |
| | Percentage of population with less than high school education | BHigh_Ratio |
| | Proportion of population in the primary, secondary, tertiary industry | PriInd_Ratio, SecInd_Ratio, TerInd_Ratio |
| | Unemployment rate | Unemp_Rate |
| | Per capita road area | Road_PC |
| | Proportion of residential, industrial land | Res_Ratio, Ind_Ratio |
| | Per capita edible oil, vegetable, red meat, fruit, edible sugar, aquatic products, egg, milk consumption | Oil, Veg, Meat, Fruit, Sugar, AquaProducts, Egg, Milk |
| **Adaptability (15)** | GDP per capita | GDP_PC |
| | Proportion of people participating in basic medical insurance | BMI_Rate |
| | Per capita disposable income of urban/rural residents | PDI_Urban, PDI_Rural |
| | Number of health technicians per thousand people | NHT_PTP |
| | Number of beds in health institutions per thousand people | NB_PTP |
| | General public budget expenditure | GPBE |
| | Number of general institutions of higher learning/high schools/all schools per thousand people | Uni_PTP, High_PTP, School_PTP |
| | Per capita park area | ParkArea_PC |
| | Proportion of green space | Green_Ratio |
| | The number of attractions/supermarkets and markets/sports venues per 100,000 people | Attractions_Ratio FreshFood_Ratio Sport_Ratio |

### 2.3. Calculation of Exposure Indexes

The land surface temperature (LST) was obtained by using a reconstruction method that combined MODIS LST with simulated temperature from the Climate Forecast System version 2 (CFSv2) [36]. The method was applied to the GEE (Google Earth Engine) platform to generate national daily kilometer grid surface temperature data for the period 2010–2020. The accuracy of the method was verified, with results showing Pearson correlation coefficients above 0.9 between the reconstructed temperature data and MODIS LST, and root mean square error and mean absolute error of about 3 °C (Table S1).

Based on the reconstructed daily land surface temperature data, the nth percentile of daily average temperature, the standard deviation of daily temperature, and the nth percentile of daily temperature difference were calculated. The nth percentile of daily mean temperature was obtained by calculating the 1st, 3rd, 10th, 25th, 50th, 75th, 90th, 97th and 99th percentile of the daily temperature series for each year. The standard deviation of daily temperatures was obtained by calculating the standard deviation of the daily surface temperature series for each year. The nth percentile of the daily temperature difference was obtained by calculating the temperature difference every two adjacent days and then calculating the 10th, 25th, 50th, 75th and 90th percentile of the annual temperature difference series.

The nth percentile of intraday–interday temperature variability (IITV) was calculated using the hourly 2 m temperature data from ECMWF ERA-5 dataset. This dataset has a spatial resolution of 0.25° × 0.25° and was resampled to a kilometer grid using the nearest neighbor sampling method. The intraday–interday temperature variability was calculated by the standard deviation of the daily minimum and maximum temperatures ($T_{min}$ and $T_{max}$, respectively) during the exposure day. The IITV of every two days was calculated by Formula (1) [37], and then determined the 10th, 25th, 50th and 75th percentile of the annual IITV series.

$$\text{IITV} = \text{SD}(T_{min}lag0,\ T_{max}lag0,\ T_{min}lag1,\ T_{max}lag1) \tag{1}$$

According to Formula (2), the above temperature exposure indicators were population-weighted by WorldPop 1 km × 1 km population grid data and the grid data were aggregated to the provincial scale. In the formula, $k$ represents the $k$th indicator; $j$ represents the region, $N$ represents the number of grids in region $j$; $i$ represents the $i$th grid; $Population^{j,i}$ represents the number of population in grid $i$ of region $j$; and $NT_k^{j,i}$ represents the kth indicator in grid $i$ of region $j$. Calculate the product of population number and corresponding indicators of all grids in region $j$ and add them together, then divide them by the total population number of the region to obtain $PNT_k^j$ which represents the population weighted indicator in region $j$.

$$PNT_k^j = \frac{\sum_i^N Population^{j,i} \times NT_k^{j,i}}{\sum_i^N Population^{j,i}} \tag{2}$$

Three meteorological exposure indexes, annual relative humidity, annual surface air pressure and annual 10 m wind speed, were obtained and calculated from ECMWF ERA-5 datasets. The nearest neighbor sampling method was used to resample them to a kilometer grid and Formula (2) was used for population weighting.

PM data were obtained from China's 1 km high-resolution and high-quality daily $PM_{2.5}/PM_{10}$ dataset of National Earth System Science Data Center. This dataset was produced from big data (ground-based observation, satellite remote sensing products, atmospheric reanalysis data, etc.) by using artificial intelligence technology and combining the spatio-temporal heterogeneity of air pollution [38–40]. Based on this dataset, the average annual $PM_{2.5}/PM_{10}$ concentration and the number of days per year when $PM_{2.5}/PM_{10}$ concentration did not reach the WHO guidelines for the three transitional periods were calculated. The provincial level PM exposure indicators were obtained by population weighting and spatial aggregation according to Formula (2).

As high-quality ground observation products for measuring $SO_2$, $NO_2$, $O_3$ and CO are scarce, only tropospheric column concentrations of these pollutants can be obtained through available remote sensing data. Therefore, we obtained the following annual average air pollutant concentrations from eco-environmental bulletins issued by ecological environment departments of each province: $SO_2$, $NO_2$, CO 24 h mean 95th percentile, and $O_3$ daily maximum 8 h moving average 90th percentile concentrations. Additionally, we determined the percentage of days in a year when the Air Quality Index (AQI) was not in good condition.

### 2.4. Calculation of Sensitivity and Adaptability Indexes

The age structure and gender ratio indicators were calculated by using WorldPop resident population age and gender structure 100 m × 100 m grid dataset. The population proportion of each age group was obtained by spatial aggregation method and age re-division. The number of attractions, supermarkets and markets per 100,000 people, and the number of sports venues per 100,000 people can be calculated by using the POI of Goldman Maps. Other indicators were obtained from corresponding data from China Statistical Yearbook, provincial statistical yearbooks and provincial statistical bulletins on national economic and social development.

### 2.5. Evaluation Method of Population Health Vulnerability

The process of population health vulnerability evaluation is illustrated in Figure 2. First, the factor detection method of geographic detector was used to analyze the correlation between each index and six kinds of chronic diseases [41]. The indexes that showed no significant correlation with the chronic disease mortality were excluded ($p > 0.05$). Then, the Pearson correlation coefficients for the remaining indicators were calculated. For the indicators with correlation coefficients exceeding 0.8, only those with the highest q-values in the factor detection were retained, while the others were excluded.

The response turning points of the selected indicators to the mortality of six kinds of diseases were determined by piecewise regression method, and the indicators were standardized on this basis. For indicators with no turning point in the positive direction, Formula (3) was used for standardization. For indicators with turning points in the positive direction, Formula (4) was used for standardization. For indicators with no turning point in the negative direction, Formula (5) was used for standardization. For indicators with turning points in the negative direction, Formula (6) was used for standardization. In the formulas, $x_1$, $x_2$, ..., $x_n$ is the original data sequence, $y_1$, $y_2$, ..., $y_n \in [0, 1]$ is the standardized data sequence, and threshold is the turning point in the correlation between indicators and corresponding diseases. The disease mortality was taken as the health result to verify the performance of the weight determination method based on quantile regression, the weight determination method combining factor detection and AHP (FAHP), and the entropy weight method in the population health vulnerability assessment. Finally, the quantile regression weight determination method with highest precision was used to assess the health vulnerability of all provinces in 2020. The vulnerability value was calculated according to Formula (7), in which V is the vulnerability value, E is the exposure score, S is the sensitivity score and A is the adaptability score.

$$y_i = \frac{x_i - \min\limits_{1 \leq j \leq n}\left\{x_j\right\}}{\max\limits_{1 \leq j \leq n}\left\{x_j\right\} - \min\limits_{1 \leq j \leq n}\left\{x_j\right\}} \tag{3}$$

$$y_i = \begin{cases} \frac{x_i - threshold}{\max\limits_{1 \leq j \leq n}\left\{x_j\right\} - threshold}, & x_i \geq threshold \\ \frac{threshold - x_i}{threshold - \min\limits_{1 \leq j \leq n}\left\{x_j\right\}}, & x_i < threshold \end{cases} \tag{4}$$

$$y_i = \frac{\max\limits_{1 \le j \le n}\{x_j\} - x_i}{\max\limits_{1 \le j \le n}\{x_j\} - \min\limits_{1 \le j \le n}\{x_j\}} \tag{5}$$

$$y_i = \begin{cases} 1 - \dfrac{x_i - threshold}{\max\limits_{1 \le j \le n}\{x_j\} - threshold}, & x_i \ge threshold \\[2ex] 1 - \dfrac{threshold - x_i}{threshold - \min\limits_{1 \le j \le n}\{x_j\}}, & x_i < threshold \end{cases} \tag{6}$$

$$V = E + S - A \tag{7}$$

To determine the weight of each index, quantile regression was used to obtain the regression coefficient of the selected indexes at the 0.05th, 0.25th, 0.5th, 0.75th, and 0.95th quantiles of the corresponding chronic diseases' mortality levels. The average regression coefficient of adjacent quantiles was calculated after excluding non-significant coefficients. The coefficients for mortality levels <0.25, 0.25–0.5, 0.5–0.75 and >0.75 were obtained and the proportion of each index coefficient in each interval was calculated as the weight for each index at that mortality level. The vulnerability values of different regions for the four mortality levels were calculated and used as features, with different mortality levels as categories. Next, a random forest model was trained and optimized using the ten-fold cross-validation method for predicting the category to which the new input data belong.

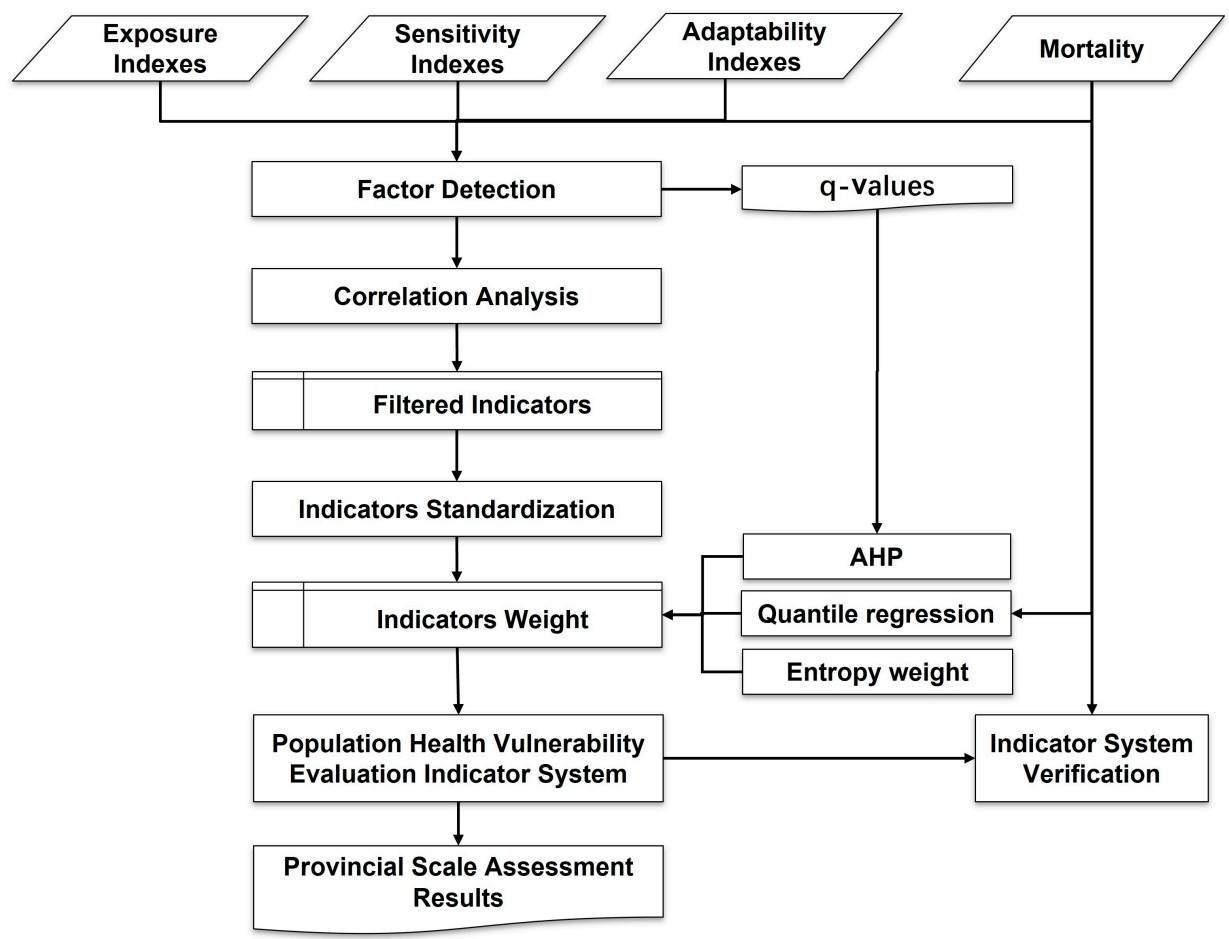

**Figure 2.** Flow chart of population health vulnerability assessment.

# 3. Results

## 3.1. Factor Detection Results of Exposure Indexes

Figure 3 shows that the low quantile temperature exposure indexes have a greater explanatory power for mortality related to respiratory diseases, heart diseases and cerebrovascular diseases than the other three types of diseases. The mortality of respiratory diseases and heart diseases can be explained by long-term temperature changes by over 60%, and the mortality of cerebrovascular diseases and malignant tumors by about 40%. Short-term temperature changes are strongly correlated with respiratory diseases, heart diseases, cerebrovascular diseases and endocrine nutrition metabolic diseases. Apart from a relative strong correlation between CO_YM and mortality related to endocrine nutrition metabolic diseases, other air pollution exposure indicators have weaker explanatory power for chronic disease mortality, with q-values around 0.2 to 0.3. Humidity exposure exhibits a strong correlation with mortality of respiratory diseases, heart diseases and endocrine nutrition metabolic diseases. Air pressure is strongly correlated with mortality of cerebrovascular diseases and digestive system diseases. Wind speed is only strongly correlated with the mortality of endocrine nutrition metabolic diseases.

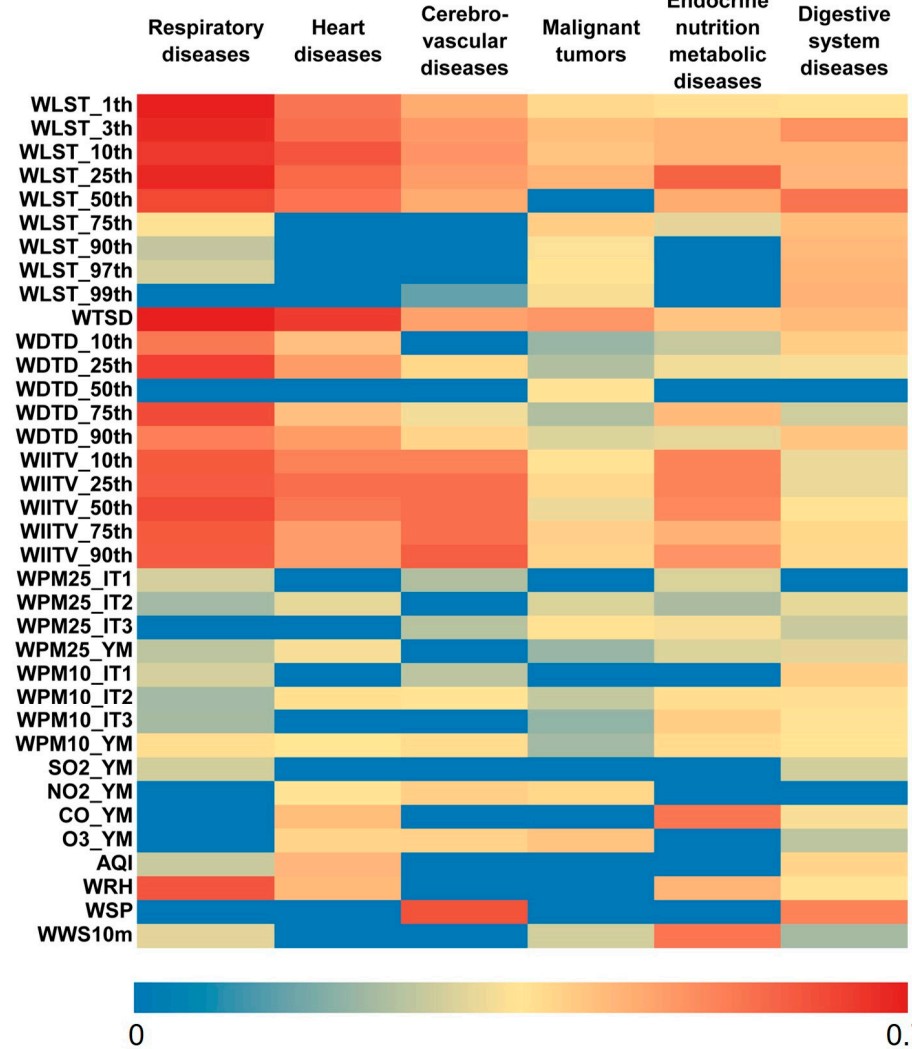

**Figure 3.** q-value distribution map of exposure indexes. The color of each grid indicates the explanatory degree of different exposure indicators to the corresponding chronic disease mortality. The dark blue indicates that there is no statistically significant relationship between the corresponding indexes and disease mortality ($p > 0.05$) while red indicates high relevance.

### 3.2. The Correlation between the Filtered Indicators and Chronic Diseases

Figure 4 demonstrates that temperature exposure is associated with mortality of each type of disease, but different temperature exposure indicators were selected for population vulnerability evaluation of different diseases. For respiratory diseases, WLST_1th and WLST_75th were selected as temperature exposure indexes, both of which are positive indicators (Table S2), indicating that high temperature exposure is positively correlated with respiratory diseases mortality. For heart and cerebrovascular diseases, long-term and short-term temperature changes were selected, indicating that the long-term and short-term temperature changes of high frequency are positively correlated with the disease mortality. Humidity is positively correlated with respiratory and digestive system disease mortality after exceeding a certain threshold (Tables S2 and S7).

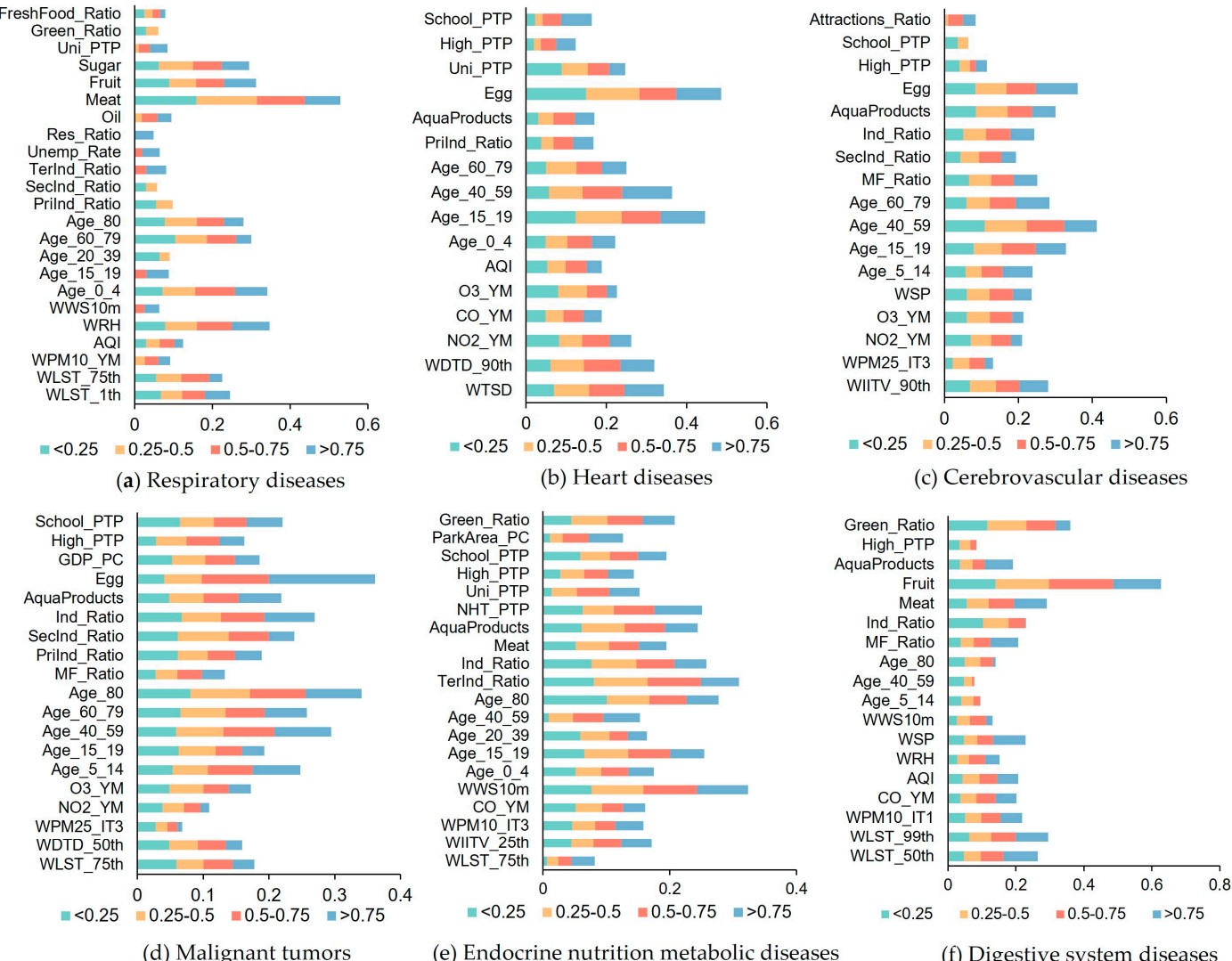

**Figure 4.** Distribution of indicators and weights obtained by quantile regression related to the six kinds of chronic diseases studied. Different colors indicate the index weights corresponding to different quantile levels (<0.25, 0.25–0.5, 0.5–0.75, >0.75) of mortality rates.

There is a correlation between age composition and mortality of chronic diseases. An increase in the proportion of infants and children is positively associated with mortality from respiratory diseases and heart diseases in some intervals (Tables S2 and S3), which may be related to the high prevalence of lower respiratory tract infectious diseases due to imperfect immune function and congenital heart defects in some infants. There is a negative correlation between the proportion of juvenile children and disease mortality, while the proportion of elderly population shows the opposite trend. In addition to respiratory diseases, the proportion of the middle-age population is positively correlated with the mortality of the other five types of disease (Tables S2–S7).

The mortality of specific diseases is related to the consumption of various foods. Per capita consumption of red meat is positively correlated with mortality from respiratory and digestive diseases, and consumption of sugar and oil is positively correlated with mortality from respiratory diseases (Tables S2 and S7). Per capita egg consumption is positively correlated with mortality from heart diseases, cerebrovascular diseases and malignant tumors (Tables S3–S5). Per capita consumption of aquatic products is positively correlated with mortality from malignant tumors, endocrine nutrition metabolic diseases and digestive system diseases (Tables S5–S7).

The mortality of different diseases is related to the proportion of population in different industries. The proportion of the population in the secondary industry is positively correlated with the mortality of respiratory diseases, cerebrovascular diseases and malignant tumors (Tables S2, S4 and S5). The proportion of the population in the primary industry has a negative correlation with the mortality of heart diseases and malignant tumors (Tables S3 and S5). Additionally, there is a positive correlation between the proportion of the population in the tertiary industry and the mortality of endocrine nutrition metabolic diseases (Table S6).

The mortality of chronic diseases is related to the proportion of different land use types and per capita educational resources. The increase in the proportion of industrial land is positively associated with mortality due to cerebrovascular diseases, malignant tumors, and endocrine nutrition metabolic diseases (Tables S4–S6). Conversely, the increase in the proportion of green space is associated with a decrease in mortality of respiratory diseases, endocrine metabolic diseases and digestive system diseases (Tables S2, S6 and S7). Per capita educational resources are negatively correlated with the mortality of chronic diseases, except for digestive system diseases.

### 3.3. Comparison of Different Weight Determination Methods

Table 2 shows the determination coefficients between vulnerability scores calculated by the three weight determination methods and the corresponding disease mortality. The quantile regression method shows the highest coefficients of determination, with $R^2$ values exceeding 0.7 for respiratory system diseases and malignant tumors. For chronic diseases excluding digestive system diseases, FAHP method obtains the second highest $R^2$ after the quantile regression method, which are very close to those of quantile regression method in cerebrovascular diseases, malignant tumors and endogenous metabolic diseases. The entropy weight method consistently shows the lowest determination coefficients, only reaching those of the other two methods in cerebrovascular diseases, malignant tumors and digestive system diseases. By comparing the index weight distribution of population vulnerability evaluation index system of various diseases in Figures 4, S1 and S2, it can be seen that the index weight distribution obtained by the quantile regression method is similar to that of the FAHP method, while the index weight distribution obtained by the entropy weight method is only comparable to the other two methods in cerebrovascular diseases, malignant tumors and digestive diseases.

**Table 2.** Determination coefficients between vulnerability calculated by different weight determination methods and corresponding disease mortality.

|  | Quantile Regression | FAHP | Entropy Weight |
|---|---|---|---|
| Respiratory diseases | 0.78 | 0.55 | 0.30 |
| Heart diseases | 0.65 | 0.57 | 0.28 |
| Cerebrovascular diseases | 0.58 | 0.58 | 0.41 |
| Malignant tumors | 0.73 | 0.72 | 0.42 |
| Endocrine nutrition metabolic diseases | 0.68 | 0.62 | 0.03 |
| Digestive system diseases | 0.66 | 0.44 | 0.51 |

*3.4. Analysis of the Results of Health Vulnerability Assessment and Main Related Factors in Different Provinces*

Figure 5 illustrates the regional vulnerability to chronic diseases in 2020. Southwest provinces showed the highest vulnerability to respiratory diseases in 2020. For heart diseases, vulnerability values exceeding 0.5 were observed in Shanghai, Shandong, northeastern provinces, and northern provinces except Shanxi. For cerebrovascular diseases, vulnerability values exceeding 0.5 were observed in Jiangsu, Shanghai, Tianjin, Shandong, and northeastern provinces. For malignant tumors, vulnerability values exceeding 0.5 were observed in Beijing, Hebei, Jiangsu, Liaoning, Shandong, Shanghai, Tianjin, Zhejiang and Chongqing. Shanghai had the highest vulnerability to endogenous metabolic diseases, whereas the western region had the lowest. The highest vulnerability to digestive system diseases was observed in Tibet, followed by Guangdong and Hainan.

Figure 6 illustrates the standardized values of meteorological exposure indexes, revealing that provinces in southwestern and southeastern China experienced more high temperatures and humid weather, increasing the vulnerability of people at risk of respiratory disease. The provinces in northeast China, north China and northwest China exhibited sharper long-term and short-term temperature changes, increasing the health vulnerability of people at risk of heart disease, cerebrovascular diseases, malignant tumors and endocrine nutrition metabolic diseases. With the exception of Gansu, Guizhou, Heilongjiang, Jilin, Liaoning, Inner Mongolia and Yunnan, all other provinces had higher WLST_75th scores, which exacerbated the health vulnerability of people at risk of malignant tumors and endocrine nutrition metabolic diseases.

In terms of air pollution exposure indicators, PM exposure in Xinjiang, Hebei, Henan, Shandong and Shanxi was severe, increasing the population's health vulnerability in these provinces. The exposure values of $NO_2$ and $O_3$ were relatively large in Hebei, Henan, Tianjin, Shanxi, Shandong, Jiangsu, Shanghai and Chongqing, which increased the population's vulnerability to heart diseases, cerebrovascular diseases, and malignant tumors. Severe CO exposure in Hebei, Shanxi, Tianjin and Liaoning increased the population's vulnerability to heart diseases, endocrine nutrition metabolic diseases and digestive system diseases.

Regarding age composition indicators, the proportion of teenagers was relatively small in northeastern provinces, Beijing, Tianjin, Inner Mongolia, Hunan, Shandong, Shanghai and Zhejiang, increasing the health vulnerability in respiratory diseases, heart diseases, cerebrovascular diseases and endocrine nutrition metabolic diseases. The higher proportion of the population aged 40–59 in the northeastern provinces, Hubei, Inner Mongolia, Shanghai and Tianjin increased the health vulnerability values in heart diseases, cerebrovascular diseases, endocrine nutrition metabolic diseases and digestive system diseases. Additionally, Jiangsu, Liaoning and Shanghai had higher Age_80 indicator scores compared to other provinces, increasing health vulnerability values in respiratory diseases, cerebrovascular diseases and endocrine nutrition metabolic diseases (Figure 7).

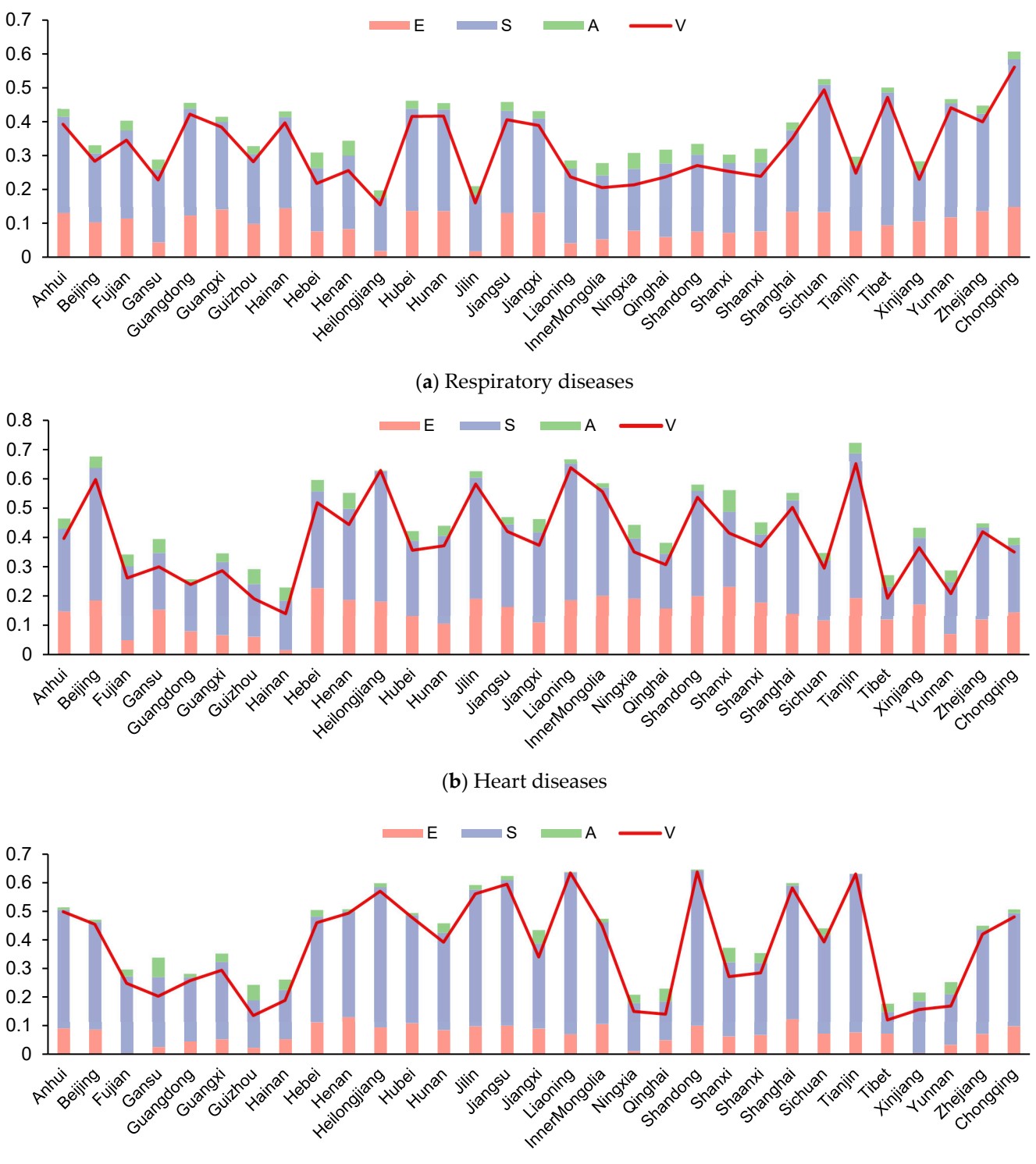

(**a**) Respiratory diseases

(**b**) Heart diseases

(**c**) Cerebrovascular diseases

**Figure 5.** *Cont.*

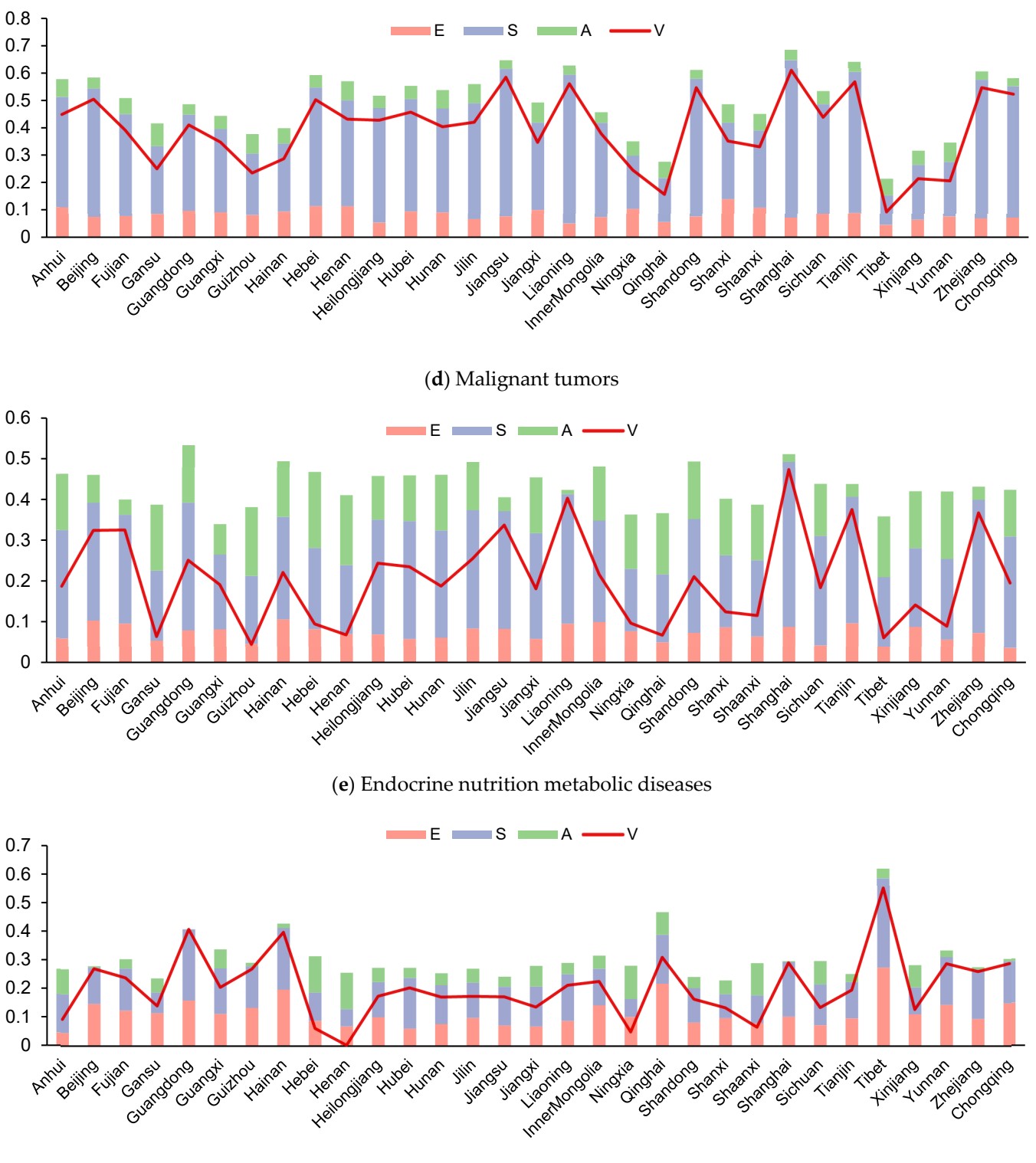

(**d**) Malignant tumors

(**e**) Endocrine nutrition metabolic diseases

(**f**) Digestive system diseases

**Figure 5.** Provincial assessment results of the population health vulnerability for the six types of chronic diseases in 2020. The different colors of the bar stack plot represent the exposure score (E), sensitivity score (S) and adaptability score (A), respectively. The red line indicates the vulnerability assessment score V. Since V = E + S − A, the points of the line do not overlap with the top of the bar chart but have a distance of value from the top.

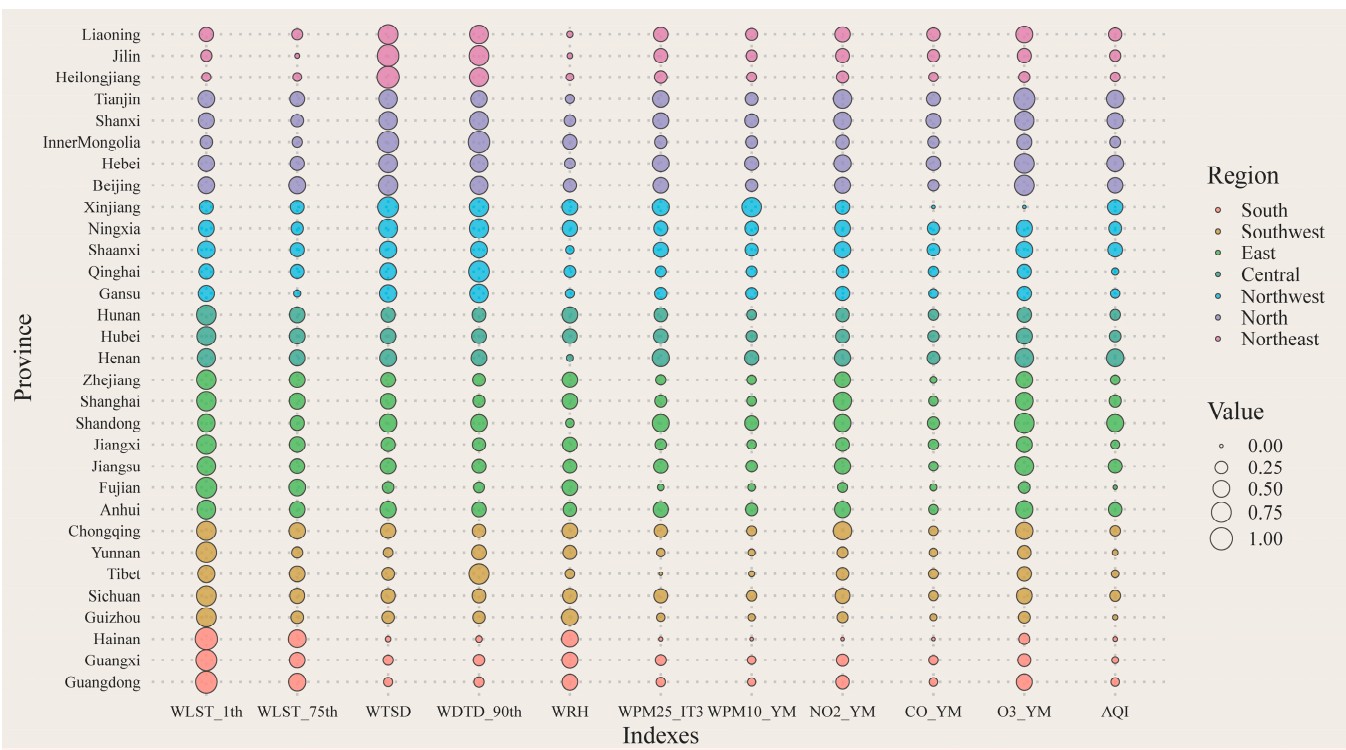

**Figure 6.** Distribution map of standardized values of main exposure indicators related to chronic diseases for each province. Each circle represents the standardized value of the corresponding exposure indicator for each province. The color of the circle indicates the region where the province is located, and the size of the circle indicates the value of the exposure index.

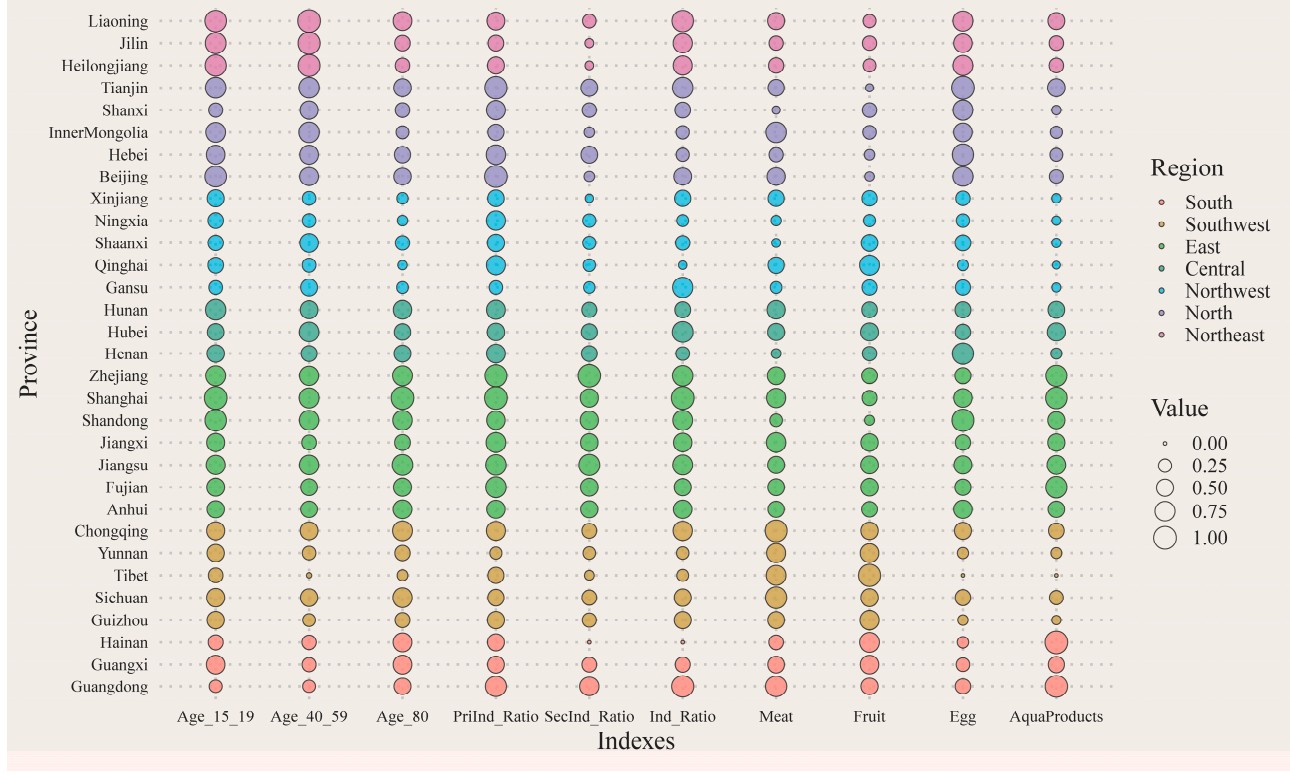

**Figure 7.** Distribution map of standardized values of main sensitivity indicators for each province.

In terms of food consumption indicators, high per capita meat consumption in Inner Mongolia, Sichuan, Tibet and Yunnan increased the health vulnerability of people to respiratory diseases, endocrine nutrition metabolic diseases and digestive system diseases. Higher per capita egg consumption in Henan, Shandong, northeastern and northern provinces increased the health vulnerability of people to heart diseases, cerebrovascular diseases and malignant tumors. Coastal provinces including Zhejiang, Shanghai, Fujian, Hainan and Guangdong had high index values of aquatic products consumption, which increased the health vulnerability of people to cerebrovascular diseases, malignant tumors, endocrine nutrition metabolic diseases and digestive system diseases (Figure 7).

Provinces with strong economic development, such as Beijing, Tianjin, Zhejiang, Shanghai and Guangdong had a low proportion of population engaged in the primary industry, which increased the health vulnerability of the population to heart diseases and malignant tumors. Guangdong, Jiangsu and Zhejiang had high SecInd_Ratio index scores due to the high proportion of population engaged in secondary industries, increasing the health vulnerability of the population to respiratory diseases, cerebrovascular diseases and malignant tumors (Figure 7).

Regarding land use indicators, the proportion of industrial land was high in Tianjin, Hubei, Shanghai, Guangdong and northeastern provinces (Figure 7), resulting in increased sensitivity scores for cerebrovascular diseases, malignant tumors, endocrine nutrition metabolic diseases and digestive system diseases. The proportion of green space was high in Hebei, Henan, Ningxia, Shanxi, Qinghai, Tibet and Guangxi (Figure 8), which increased the adaptability to respiratory system diseases, endocrine nutrition metabolic diseases and digestive system diseases. However, Beijing, Hubei, Shanghai and Guangdong had lower proportions of green space compared to other provinces.

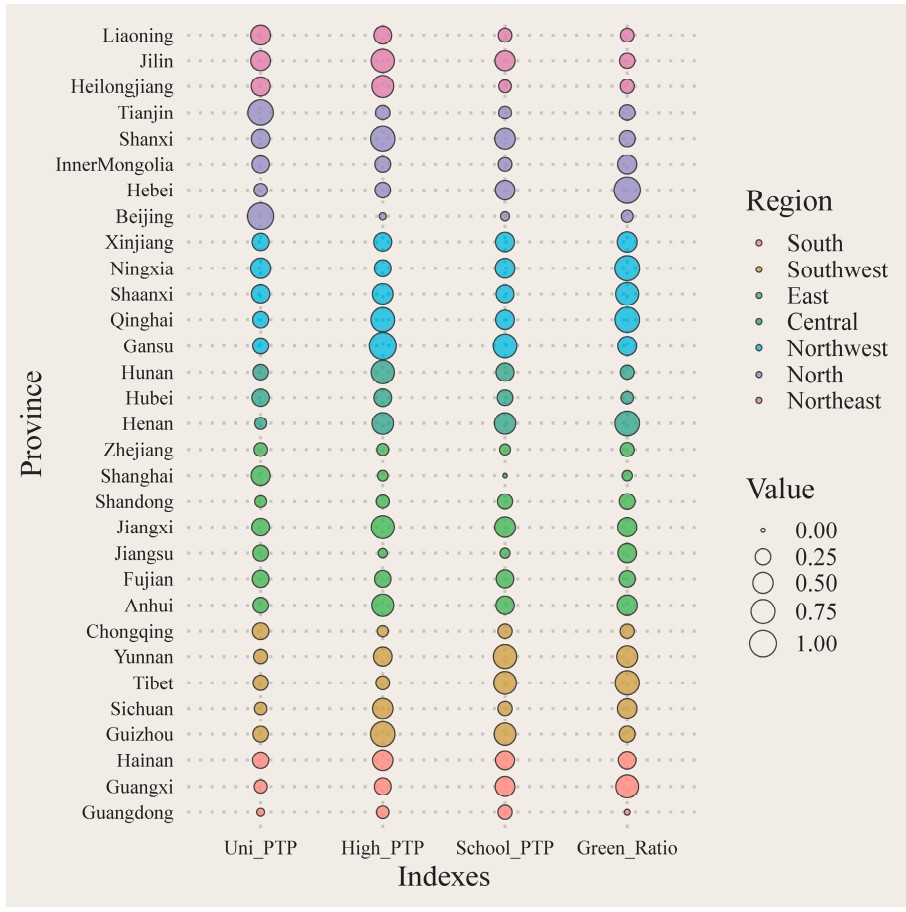

**Figure 8.** Distribution map of standardized values of main adaptability indicators for each province.

Per capita education resources play an important role in increasing adaptability to chronic diseases. Beijing and Tianjin had a much higher number of general higher education schools per thousand people compared to other provinces, while densely populated provinces such as Zhejiang, Shanghai, Jiangsu, Chongqing and Guangdong had lower numbers of secondary schools and schools per thousand people (Figure 8). This indicates a potential reduction in adaptability to chronic diseases in these densely populated provinces.

## 4. Discussion

### 4.1. Estimation Error of Air Pollution Exposure

The results of factor detection show that there is not a strong correlation between air pollution exposure indicators and mortality rate for various diseases. However, numerous studies have shown significant impacts of air pollution on respiratory, cardiovascular and cerebrovascular diseases [26,30,31,42–48]. This may be due to errors in air pollution exposure estimation caused by rapid changes in air quality and population mobility. Although the daily PM distribution data with high temporal resolution were used in this study, some heavily polluted areas (e.g., Beijing, Tianjin, Hebei) may experience abrupt spikes in $PM_{2.5}$ concentrations [49]. Although this study utilized high spatial resolution annual population distribution data for air pollution exposure indicator calculations, the limitation of data acquisition may result in errors due to cross-regional population movement at different times not being accounted for. Some studies have shown that using mobile phone location data rather than fixed population data to estimate air pollution exposure will provide more accurate results [50].

In addition, failure to obtain indoor air pollution data can also lead to errors in estimation. Studies on human exposure to indoor pollution have shown that indoor environment is at least twice as polluted as the outdoor environment. A recent report concluded that household air pollution is a major contributor to global morbidity and mortality, with significant effects on the respiratory and cardiovascular systems [51].

### 4.2. Explanation of the Influence of Main Sensitivity and Adaptability Indexes on the Health Vulnerability of People

The screening results of the indexes reveal a significant correlation between per capita consumption of various food and chronic diseases. Among them, the consumption of aquatic products, red meat and eggs has been linked to increased population health vulnerability. In the case of aquatic products, it may because heavy metal contamination in water bodies has gradually become a serious problem since China's rapid industrialization and urbanization [52,53]. Aquatic products will accumulate heavy metals from the environment, which move up the food chain, and consumption of contaminated products poses a potential risk to human health [54]. Wang et al. analyzed the spatial distribution of heavy metals in four types of aquatic products from 32 provinces during 2015–2017. The results show that the cancer risk caused by cadmium or chromium cannot be ignored [55]. For red meat, some studies report that the risk of dying from cancer, heart disease, respiratory disease, stroke, diabetes, infection, kidney disease or liver disease increases with the consumption of red meat [56]. In addition, the effect of nitrites make processed meat associated with increased cardiovascular and respiratory mortality [57]. Many provinces in southern China have the habit of making and eating bacon. Increased egg intake may lead to an increased risk of cardiovascular disease and cancer death [58]. In addition, studies have shown that egg intake is a risk factor for colon, rectal and prostate cancer [59].

In addition to food consumption, food processing methods can also have an impact on human health. For example, high sodium and low potassium diets can increase blood pressure and ultimately cardiovascular disease. In China, the average dietary sodium content is excessively high and potassium content is insufficient. However, there is a lack of accurate data on sodium and potassium intake. Tan et al. used meta-analysis to summarize all published 24 h urinary sodium and potassium data to determine provincial sodium and potassium intake [60]. The study showed higher sodium intake in Tibet, Ningxia,

Hebei, Henan, Shandong, Jiangsu, Heilongjiang, Liaoning, Beijing and Tianjin, which is consistent with the conclusion of this study that population in northeastern provinces, northern provinces and Shandong have higher health vulnerability values to heart diseases.

The proportion of population in different industries is related to the mortality of different diseases. The proportion of the secondary industry population is positively correlated with the mortality of respiratory diseases, cerebrovascular diseases and malignant tumors. Secondary industry refers to mining, manufacturing, power (heat, gas and water production and supply), construction and similar businesses. The working environment of these industries is mostly accompanied by air pollution and chemicals which are harmful to population health. The proportion of population in the primary industry has a negative correlation with the mortality of heart diseases and malignant tumors. The primary industry mostly encompasses agricultural practices such as forestry, animal husbandry, and fishing, which typically involve prolonged periods of physical activity. Research has shown that increasing exercise can decrease the incidence of obesity and high blood pressure, ultimately reducing the morbidity and mortality associated with cardiovascular disease and other related conditions [61]. At the same time, production activities associated with the primary industry are predominantly located in rural areas and are subject to fewer impacts from human activities. Consequently, air pollution levels in these areas are relatively low, and the heat island effect is less pronounced than in urban areas. As a result, individuals engaged in primary industry work may experience relatively limited environmental exposure. There is a positive correlation between the proportion of population in the tertiary industry and endocrine nutrition metabolic diseases mortality, which may be due to less physical activity and higher stress levels associated with this sector, which can increase the risk of developing endocrine and metabolic diseases such as obesity and diabetes.

The study results show that the number of public education facilities per capita helps increase the adaptability to chronic diseases. This could be attributed to the fact that the number of public education facilities per capita reflects the proportion and education level of educated people in a certain area to some extent, who possess better health awareness and knowledge about how to mitigate risk factors, ultimately leading to better adaptability to environmental exposure.

*4.3. Influential Factors of Chronic Diseases Not Considered in This Study*

Smoking and drinking are two factors that have been widely proved to have a negative impact on the occurrence and development of various diseases. Smoking is a risk factor for cardiovascular diseases, diabetes, cancer and other diseases, and exposure to second-hand smoke can increase the incidence and mortality rate of cardiovascular diseases [62]. Alcohol use can increase the risk of cardiovascular disease from different dimensions, and plays a causal role in the development of oral cavity, pharynx, larynx, esophagus, liver, colon, rectal and female breast cancers, and may be associated with gastric and pancreatic cancers [63]. This study was unable to include indicators on tobacco and alcohol consumption for each province due to lack of historical data. If this data can be supplemented in future studies, it may improve the accuracy of population health vulnerability assessment. Moreover, the COVID-19 pandemic that emerged in late 2019 has had an impact on the mortality of various chronic diseases, which may persist for years to come. Therefore, future research should take into account the impact of such public health emergencies on chronic mortality rates.

## 5. Conclusions

This study took the advantage of remote sensing data to finely characterize meteorological factor exposures and air pollution exposure, and constructed population health vulnerability evaluation index systems for six types of chronic diseases based on an exposure–sensitivity–adaptability framework. The validation of the study's results through mortality data and previous research demonstrates the effectiveness of the proposed method in assessing population health vulnerability regarding non-communicable disease mortality risk at the regional level.

Factor detection results indicate that low quantile temperature exposure indexes and temperature change indexes exhibit relatively strong explanatory power for chronic disease mortality. In contrast, most air pollution exposure indicators demonstrate weaker explanatory power for chronic disease mortality, with q-values ranging from 0.2 to 0.3.

The results of the population health vulnerability assessment for each province in 2020 show that the southwestern provinces had the highest vulnerability values for respiratory diseases. The northeastern provinces, the northern provinces, Shanghai and Shandong had high vulnerability values for heart diseases. Jiangsu, Shanghai, Tianjin, Shandong and the northeastern provinces had high vulnerability values for cerebrovascular diseases. Beijing, Hebei, Jiangsu, Liaoning, Shandong, Shanghai, Tianjin, Zhejiang and Chongqing had higher malignant tumor vulnerability values. Shanghai had the highest vulnerability values for endocrine nutrition metabolic diseases, while the western regions had lower ones. Tibet had the highest vulnerability value for digestive system diseases, followed by Guangdong and Hainan.

Temperature exposure, air pollution exposure, age structure, food consumption, proportion of population in different industries, land use and number of public education facilities per capita are the main factors related to mortality of chronic diseases. The population in southern region should pay attention to the exposure of high-temperature and humid weather, and the population in northeastern, northern and northwestern provinces should pay attention to the impact of temperature change. The air pollution exposure in Xinjiang, Hebei, Henan, Shandong and Shanxi is more serious. The population aging level in Tianjin, Inner Mongolia, Shanghai and the northeastern provinces is relatively high, so the health support system for the elderly should be improved as soon as possible. In addition, the results show that not only people over 60 years old are vulnerable to health, but people aged 40–59 years old also have a strong sensitivity to a variety of chronic diseases which may be due to the high pressure of upbringing and supporting families. Therefore, attention should also be paid to disease prevention for middle-aged people.

The mortality rate of chronic diseases is affected by various factors such as natural environment, population characteristics and regional socio-economic conditions, and the form of their impact is complex. Currently, there is no unified indicator system for evaluating the relevant factors of chronic diseases in existing research. In this study, using remote sensing and geographic information system technology, we constructed indicator systems for chronic diseases and analyzed the relevant factors. Based on this, we obtained preliminary results of the assessment of population health vulnerability nationwide, which can contribute to the formulation of regional environmental governance and social security policies. Because the urban environment is highly variable, intervention at the overall level is more cost-effective than intervention at the individual level. Therefore, studies on how the environment affects chronic diseases are of great significance to protect residents' health and reduce the national financial expenditure on medical care. At the same time, studies on the environmental exposure and coping ability of the population at a holistic level may provide a new perspective on prevention and control of chronic diseases. However, there are still some issues that need further research and exploration.

Firstly, the accuracy of the patterns obtained from exploring the correlation between various factors and chronic disease mortality is influenced by the completeness of mortality data. In this study, we analyzed chronic disease mortality data from 2010 to 2020, but some provinces had missing mortality data in certain years. In future studies, obtaining more complete and longer-term mortality data for chronic diseases would aid in identifying more accurate relationships between environmental factors and chronic disease mortality responses, while improving the precision of the evaluation model. Furthermore, utilizing the method of this study to conduct more detailed studies at finer scales such as cities, county-level and communities based on more comprehensive chronic disease mortality data can help to investigate the scale effect of various factors on chronic disease mortality.

Secondly, a comprehensive and accurate characterization of the factors associated with chronic disease mortality is essential for assessing the health vulnerability of the

population. However, there is still room for improvement in both aspects of this study. With regards to the comprehensiveness of the indicators, as discussed in Sections 4.1 and 4.3, this study was unable to quantify certain indicators that affect chronic disease mortality, such as regional tobacco and alcohol consumption and indoor pollution, due to limitations in data acquisition. In addition, chronic diseases are greatly influenced by genetics, but there is currently no authoritative and easy-to-implement parameterization method for this aspect. These are research directions for optimizing the evaluation system. In terms of the accuracy of indicators, the general public budget expenditure in sensitive indicators can partially reflect the economic development of a region. However, more specific public budget expenditure indicators in healthcare, social security and education may provide stronger insights into the factors that influence chronic disease mortality. The analysis in Section 4.1 on exposure error in air pollution also suggests that obtaining air pollution monitoring data with higher temporal resolution and population mobility data can aid in more accurate estimation of environmental exposure. This, in turn, can lead to more precise relationships between environmental exposure and chronic disease mortality.

Exploring the factors that impact chronic disease mortality involves knowledge from various disciplines and fields. It is the correlation between each element and chronic disease mortality that was explored through the construction of indicators in this study, indicating that these indicators may not directly affect chronic disease mortality. Therefore, in future studies, it would be beneficial to integrate knowledge from various fields such as medicine, to connect the selected indicators in this study with other relevant factors. This would aid in identifying the factors that have an impact on chronic disease mortality and help in revealing the mechanisms by which each element affects chronic disease mortality.

**Supplementary Materials:** The following supporting information can be downloaded at: https://www.mdpi.com/article/10.3390/ijgi12040155/s1, Figure S1: Distribution of indicators and corresponding weights obtained by FAHP related to the six kinds of chronic diseases; Figure S2: Distribution of indicators and corresponding weights obtained by the entropy weight method related to the six kinds of chronic diseases; Table S1: Accuracy Evaluation of Land Surface temperature Reconstruction in China from 2010; Table S2: Indicators related to respiratory diseases; Table S3: Indicators related to heart diseases; Table S4: Indicators related to cerebrovascular diseases; Table S5: Indicators related to malignant tumors; Table S6: Indicators related to endocrine nutrition metabolic diseases; Table S7: Indicators related to digestive system diseases.

**Author Contributions:** Conceptualization, Chunxiang Cao; methodology, Min Xu; software, Zhibin Huang; validation, Xinwei Yang; formal analysis, Zhibin Huang; investigation, Min Xu; resources, Zhibin Huang; data curation, Zhibin Huang; writing—original draft preparation, Zhibin Huang; writing—review & editing, Min Xu; visualization, Zhibin Huang; supervision, Chunxiang Cao; project administration, Chunxiang Cao; funding acquisition, Chunxiang Cao and Xinwei Yang. All authors have read and agreed to the published version of the manuscript.

**Funding:** This research was funded by the National Key R&D Program of China, grant number 2021YFC1523503.

**Data Availability Statement:** The data presented in this study are available on request from the corresponding author. PM data presented in this study are not publicly available because the data were accessed on 20 July 2022 from http://geodata.nnu.edu.cn/.

**Acknowledgments:** Acknowledgement for the data support from "Yangtze River Delta Science Data Center, National Earth System Science Data Center, National Science & Technology Infrastructure of China. (http://geodata.nnu.edu.cn/, accessed on 20 July 2022)".

**Conflicts of Interest:** The authors declare no conflict of interest.

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
