# Peer review of "Impact of Environmental Exposure on Chronic Diseases in China and Assessment of Population Health Vulnerability"

_ijgi, doi:10.3390/ijgi12040155_

Round 1

Reviewer 1 Report

This manuscript is written well and I answered all questions in review.

I recommend the authors add two sections: 1. Scope & Limitations of their study, e.g., data sample size, # of variables, descriptive v. predictive analysis, etc. 2. Future research - either in Conclusions or a separate section - what can be done in the future to extend this research. Some implications of their research can also be discussed.

Reviewer 2 Report

Impact of Environmental Exposure on Chronic Diseases in China and Assessment of Population Health Vulnerability:

·        Add some of the most important quantitative results to the Abstract.

·        Add/Replace the name of the study area with the Keywords.

·        In the last paragraph of the Introduction, the authors should mention the weak point of former works (identification of the gaps) and describe the novelties of the current investigation to justify that the paper deserves to be published in this journal.

·        Discuss the most important reasons for the variations of the distribution of indicators and weights obtained by quantile regression related to the six kinds of chronic diseases.

·        “The proportion of population in the primary industry has a negative correlation with the mortality of heart diseases and malignant tumors, which may be due to the fact that people engaged in the primary industry have more manual labor and work in less polluted natural environment.”. Explain.

·        At the end of the manuscript, explain the implications and future works considering the outputs of the current study.

·        The quality of the language needs to be improved for grammatical style and word use.

Reviewer 3 Report

Dear Authors,

Congratulations on your research. I really appreciate the thoroughly prepared content of the paper, however there are some issues that could be improved. 

1. 'Lines 64 - 68 - You write: ‘For example, the data of meteorological stations and air quality monitoring stations used in many studies only represent a single location and its surrounding affected areas. The distribution of stations in most areas is not uniform, so the spatial data obtained by interpolation methods are likely to have an uneven spatial distribution pattern where sites are sparse. In addition, the cohort and 68 sampling survey methods adopted by most epidemiological studies are time-consuming 69 and laborious.'  – have you searched for other studies that included preparation of a survey among inhabitants? There is literature basing on inhabitants' surveys. It would be worth including such research (for example research on the perception of the community concerning air quality or other examples of the above mentioned approach).

2. Lines 258 – 261, 375 – 378, 403 - 406 – the formatting has to be changed;

3. Lines 479 - 482 - You write 'The results of factor detection show that the correlation between air pollution exposure indicators and mortality of diseases is not high, but many short-term studies have shown significant effects of air pollution on respiratory, cardiovascular and cerebrovascular diseases' - what studies do you mean? Give some examples, providing references to the literature;

4. Lines 563-566 - 'This study took the advantage of remote sensing data to finely characterize meteorological factors exposure and air pollution exposure, and constructed population health vulnerability evaluation index systems for six types of chronic diseases by using the factor detection method combined with correlation analysis to screen the indicators.' - the aspect of the application of remote sensing data has been marginalized. The methodology should be adequately described; You could include some more details concerning the process of the analysis of the images; Have any other researchers conducted studies concerning the subject based on remote sensing?

Thank you in advance for including the remarks.

Best wishes,

Reviewer

Round 2

Reviewer 1 Report

Authors have addressed the concerns.

Reviewer 2 Report

I appreciate the authors addressing the comments. The manuscript can be accepted in its current format. Congrats!